# Pixel-level Bayer-type colour router based on metasurfaces

Xiujuan Zou[1,5], Youming Zhang [ORCID][2,5 ✉], Ruoyu Lin[1], Guangxing Gong[1], Shuming Wang [ORCID][1,3,4 ✉], Shining Zhu [ORCID][1,3,4] & Zhenlin Wang [ORCID][1,4 ✉]

The three primary colour model, i.e., red, green, and blue model, based on the colour perception of the human eye, has been widely used in colour imaging. The most common approach for obtaining colour information is to use a Bayer colour filter, which filters colour light with four pixels of an imaging sensor to form an effective colour pixel. However, its energy utilization efficiency and colour collection efficiency are limited to a low level due to the three-channel filtering nature. Here, by employing an inverse-design method, we demonstrate a pixel-level metasurface-based Bayer-type colour router that presents peak colour collection efficiencies of 58%, 59%, and 49% for red, green and blue light, and an average energy utilization efficiency as high as 84% over the visible region (400 nm–700 nm), which is twice as high as that of a commercial Bayer colour filter. Furthermore, by using a 200 μm × 200 μm metasurface-based colour router sample working with a monochromatic imaging sensor, colour imaging is further realized, obtaining an image intensity twice that achieved by a commercial Bayer colour filter. Our work innovates the mechanism of high-efficiency spectrum information acquisition, which is expected to have promising applications in the development of next-generation imaging systems.

[1] National Laboratory of Solid State Microstructures, School of Physics, Nanjing University, Nanjing 210093, China. [2] Huawei Technologies Co., Ltd, Shenzhen 518129, China. [3] Key Laboratory of Intelligent Optical Sensing and Manipulation, Ministry of Education, Nanjing University, Nanjing 210093, China. [4] Collaborative Innovation Center of Advanced Microstructures, Nanjing 210093, China. [5]These authors contributed equally: Xiujuan Zou, Youming Zhang. ✉email: zhangyouming3@huawei.com; wangshuming@nju.edu.cn; zlwang@nju.edu.cn

Most of the imaging systems employed in digital cameras and mobile phones involve a finely cascaded series of bulk optical devices and electronic devices. To obtain the colour information of the colourful world, colour imaging sensors need to be introduced into the imaging systems. However, all imaging sensors can only obtain intensity information, i.e., monochromic imaging sensors. To extract the colour information, a colour filter (CF) must be employed to allow narrow-band light to pass through and reach the imaging sensors and light of other wavelengths to be absorbed or reflected. To date, most commercialised colour imaging sensors are based on the Bayer colour filters (BCFs) invented by Bryce E. Bayer in 1976 (Fig. 1a)[1], in which each effective colour pixel is composed of four pixels covered by one red (R), two green (G) and one blue (B) filters, i.e., the RGGB pattern. Despite their widespread use, these BCFs suffer from the intrinsic energy utilisation efficiency limit due to the multichannel spectrum extraction configuration, which is a result of the tradeoff between the number of spectral channels and energy utilisation efficiency[2]. For a BCF, the ideal energy utilisation efficiency defined as the ratio of the light intensity detected by a single colour pixel to that of the total incident light is limited to 1/3, for an incident white light with the identical intensity in the visible region and the actual loss of the CFs themselves are not considered. Aiming to improve the transmittance or colour accuracy, new CFs based on various nanophotonic structures by using metallic plasmonic structures or photonic crystal structures[3–9] were subsequently proposed. However, the underlying multichannel spectrum extraction mechanism remains the same as that introduced 40 years ago, resulting in the bottleneck problem of low energy utilisation efficiency in colour imaging.

To solve this problem, some innovative solutions, including stacked colour dependent photodiodes without CFs[10–12], improved CF combinations[13] and a variety of colour routers (CRs)[14–19], have been proposed and demonstrated in recent years. Stacked colour dependent photodiodes are new types of colour imaging sensors based on organic photoconductive films to improve the light utilisation efficiency, but the large pixel size and complex organic photoelectric conversion modules are still challenging to widely apply in colour video cameras. Improved CF combinations, such as RYYB and RGBW, are applied in the market that absolutely increases the amount of light entering but simultaneously creates another problem of imaging colour cast, which requires some colour postprocessing through AI algorithms. Additionally, the CRs reported can fully utilise the light incident on the imaging sensors and route different colours to the corresponding pixels. Three-dimensional (3D) CRs have been demonstrated with the aim of improving the energy utilisation efficiency and colour collection efficiencies of RGB light by using multilayer stacked structures or 3D structures[14,15] designed using the inverse-design method[20–23]. Despite the theoretically great performance, the difficulties in fabricating 3D nanostructures in multiple lithographies and precise alignment hinder their realisation, especially for devices working in the visible region. A practical candidate for a CR is a metasurface with a wavelength-scale thickness that can arbitrarily manipulate the phase, polarisation, intensity, and other parameters of light[24,25]. By using metasurfaces, a variety of light manipulation functions, such as high-quality focusing and imaging[26–30], holography[31–35], polarisation generation[36–40], and beam steering[41–45], have been fulfilled. Taking full advantage of a metalens in precise control of light focusing, the CR function has been realised by a single-layer metalens with unit cells composed of different nanostructures corresponding to different wavelengths[16]. The use of the Pancharatnam-Berry phase metalens leads to a 1/2 limit of the energy utilisation efficiency because of the polarisation requirement as well as a much larger size of the CR based on the metalens than that of the commercial imaging sensor pixel, which is only approximately on the one-micrometre scale. By using the spectrum splitting mechanism of specially designed nanostructures, different wavelengths can be separately scattered to different positions in one direction[17]. However, either the asymmetric CR patterns in the imaging sensor plane or the resulting colours being not the three primary colours greatly compromise the colour imaging.

In this work, we propose a metasurface-based Bayer-type colour router (MBCR) suitable for a commercial complementary metal-oxide-semiconductor (CMOS) imaging sensor with a pixel size of $1\,\mu m \times 1\,\mu m$ by using the inverse-design method of a genetic optimisation. Different from the colour scattering mechanism in one direction, we demonstrate that the MBCR is suitable for the standard Bayer-type arrangement (RGGB) based on a single-layer metasurface. Our MBCR can provide significant improvement (84%) in the energy utilisation efficiency and colour collection efficiencies (~50%) of RGB light. Furthermore, full-colour imaging is realised using the fabricated MBCR, and the obtained image is demonstrated to show a higher intensity than that obtained by a commercial imaging sensor.

## Results

**Comparison of design principles between the colour router and colour filter.** Figure 1 schematically shows the imaging systems of the common BCF and the proposed MBCR. Note that

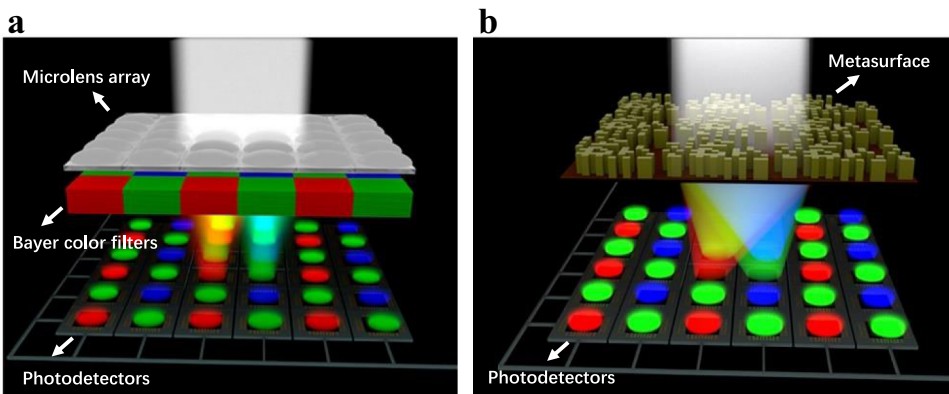

**Fig. 1 Comparison between the filtering principles of traditional BCF and the routing principle of the MBCR. a** Schematic of the traditional imaging system with BCF, microlens array and CMOS imaging sensor. **b** Schematic of the imaging system with MBCR and CMOS imaging sensor.

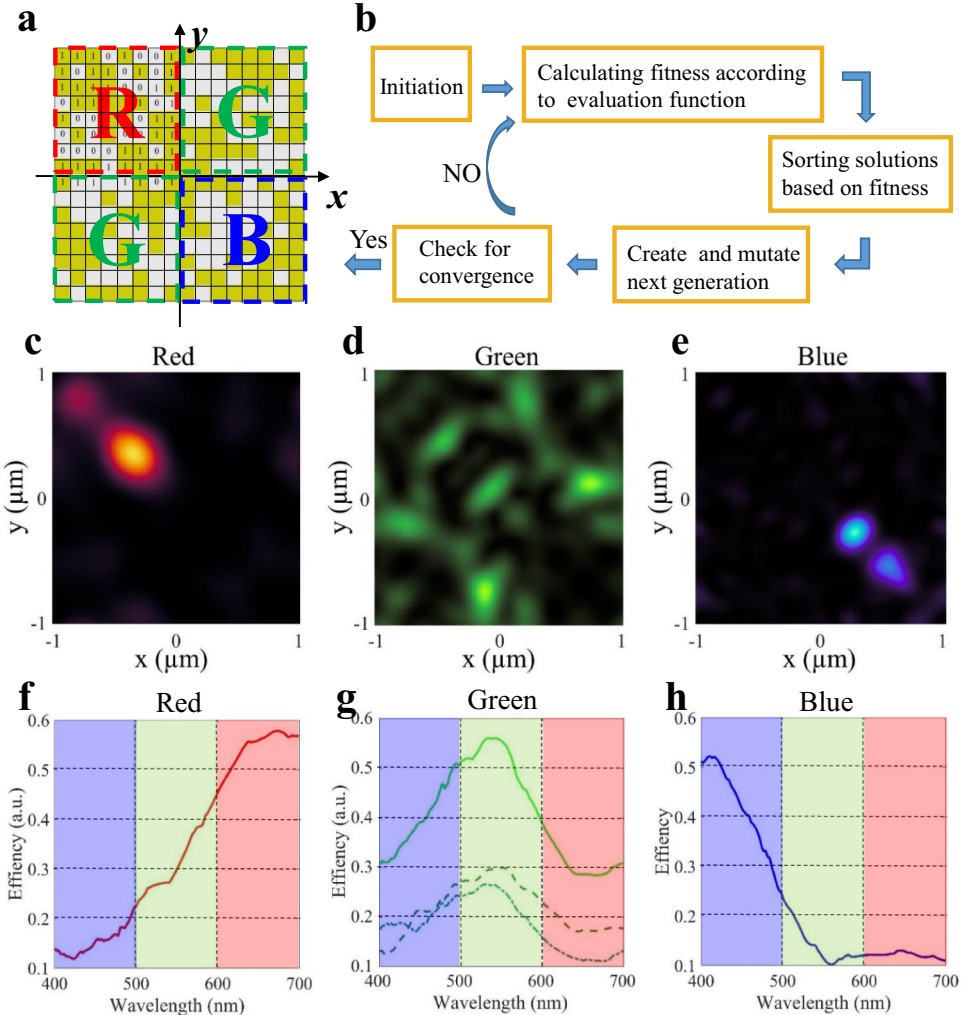

**Fig. 2 The main design process and numerical simulation results. a** Schematic of the prototype of binary-pattern MBCR. R, G and B represent the red, green and blue pixel area, respectively. **b** Flow diagram of the inverse-design progress with GA algorithm. **c–e** The calculated Poynting vector distributions of the transmitted light on the working plane for a unit-cell structure at 650, 540 and 450 nm, respectively. **f–h** The calculated optical transmittances in the visible spectrum of the four quadrants of the unit cell in MBCR. **f**, **h** correspond to the red and blue quadrant, respectively. In (**g**), two dotted curves correspond to the two green quadrants, while the solid line refers to the total energy collection efficiency of the two quadrants.

the traditional imaging system contains a BCF, a microlens array and a CMOS imaging sensor, while the imaging system we design is only composed of an MBCR and a CMOS imaging sensor. The microlens above the CF in the traditional imaging system is used to focus the incident light with a certain angle range onto the CF and is necessary for enhancing the light collecting efficiency[16]. Here, to obtain higher energy utilisation efficiency, colour collection efficiency, and device compactness, the routing property of the MBCR is adopted to improve the filtering of CFs and replace the focusing of the microlens in the common BCF. Among the popular inverse-design schemes, we choose the genetic algorithm (GA), which is a typical global optimisation algorithm for stochastic search based on the concepts of evolution and natural selection[46,47].

**Optimisation process of metasurface-based Bayer-type colour router**. A prototype of our design generated from the arbitrary initialisation is shown in Fig. 2a, which is one unit cell of the metasurface in the perpendicular direction corresponding to four pixels (marked by different coloured dotted lines). The unit cell of the MBCR has a period of 2 μm × 2 μm, matching the common pixel size (1 μm × 1 μm) of the commercial CMOS imaging sensor. In the optimisation, the unit cell is divided into 16 × 16 square

pillars with a size of $a = 125$ nm, considering the balance between the minimal feature size in fabrication and the largest parameter space for design. Each square nanopillar is assigned to either silicon nitride (Si$_3$N$_4$) or air, represented by '1' and '0', respectively, and the height of the pillar is 600 nm for optimised transmittance. The parameters of the Si$_3$N$_4$ nanopillars used for optimisation are experimentally measured, which are provided in Supplementary Fig. 1. Since the RGGB pattern has a diagonal symmetry and the MBCR must be polarisation-independent, the unit cell must be set to be diagonally symmetric in the design.

The flow diagram of the basic procedures of the optimisation algorithm is shown in Fig. 2b. First, the binary-pattern metasurface is encoded into the chromosome with binary numbers defined as either material or free space. Second, to enable GA optimisation towards the optimal result, the fitness function that evaluates the incident light routing property for each binary-pattern metasurface is defined as

$$F = a_R \int_{\lambda_{R1}}^{\lambda_{R2}} T_R d\lambda + a_G \int_{\lambda_{G1}}^{\lambda_{G2}} T_G d\lambda + a_B \int_{\lambda_{B1}}^{\lambda_{B2}} T_B d\lambda, \quad (1)$$

where $\lambda_{C1}$ and $\lambda_{C2}$ are used to represent the minimum and maximum wavelengths and the subscript C represents R, G and B

colour light. The minimum and maximum wavelengths are 600 and 700 nm for R, 500 and 600 nm for G and 400 and 500 nm for B. In the above formula, $T_R, T_G, T_B$ are the ratios of the light intensity of each colour region at the imaging plane to the light intensity incident on the unit cell, and $a_R, a_G, a_B$ are the weights of the integral for each colour. Thus, the design problem of the binary-pattern metasurface is formulated as maximisation of the fitness function $F$. The fitness function aims to maximise the specific colour light energy (waveband range: $\lambda_{C1} \sim \lambda_{C2}$) within different colour pixel areas, which is a routing process that makes light of different wavebands be directionally transmitted to each pixel. The GA is employed in combination with finite-difference time-domain simulations (Lumerical FDTD solutions) to optimise the arrangement of the meta-atoms of the metasurface. Periodic boundary conditions are applied around the unit cell in the simulation using FDTD software. The fitness value $F$ is then calculated according to the extracted transmission spectra for each binary-pattern nanostructure obtained from FDTD and is further evaluated to determine whether it has reached the target value. This optimisation process continues until the GA stop criterion is met. Here, the stop criterion depends on the mean value of the fitness function $F$ of all individuals in the current generation reaching the maximum (i.e., the evaluation function values are saturated with an increased number of iterations). More details about the implementation of the GA can be found in the Design and Simulation section of Methods and some intermediate results during the optimisation process (Supplementary Fig. 10) are provided in Supplementary Note 2. Some hyperparameters, as well as the computer resources of the GA, are also provided in Supplementary Note 2.

**Numerical simulation of the metasurface-based colour router.** Figure 2c–e show the calculated power flow density distributions for the designed MBCR at wavelengths of 660, 540, and 450 nm in the imaging plane. The majority of RGB light is routed to the desired regions. A small part of green light is scattered to red and blue regions, leading to some amount of crosstalk. However, since most of the green light energy is routed to the desired region, this crosstalk in the colour imaging can be neglected, which can be seen from the experimental imaging results shown below. The calculated colour collection efficiencies of the four quadrants in the unit cell corresponding to RGB channels in the visible spectrum are plotted in Fig. 2f–h. Figure 2f, h correspond to the red and blue quadrants, respectively. The two dotted curves in Fig. 2g correspond to the two green quadrants, while the solid line refers to the total energy collection efficiency of the two quadrants. Here, the colour collection efficiency is defined as the ratio of the power collected in each colour quadrant to the total power illuminated on the unit cell for the light of a certain wavelength. The collection efficiencies of the MBCR present a better performance than those of the commercial BCFs; the values for the latter are approximately 20–40%, as shown in Supplementary Fig. 6. In addition, the energy utilisation efficiency for the MBCR, i.e., the summation of the colour collection efficiencies of all channels, presents an average value as high as 93% (Supplementary Fig. 7) over the visible region, confirming that most of the incident light can be collected by the imaging sensor. The polarisation independence of the MBCR is also demonstrated in Supplementary Fig. 8. Although the MBCR is designed to route visible light at normal incidence, the routing performance presents robustness under incident angles up to ~30°, as shown in Supplementary Fig. 9. From the simulation data, with increasing incident angle, the peak values of the colour collection efficiencies can be analysed to accordingly slightly decrease. Moreover, we also add the simulation results under larger incident angles (40° and 50°). In

this case, serious crosstalk occurs between RGB pixels and the colour routing function cannot be realised.

**Experimental verification of the metasurface-based colour router.** The MBCR structure pattern optimised with the GA is shown in Fig. 3a. The $Si_3N_4$ nanopillars are fabricated on a quartz glass substrate with a thickness of 500 μm. Figure 3c shows a top-view scanning electron microscopy (SEM) image of a fabricated MBCR sample, which presents the fine nanostructures consistent with the optimisation design proven in Supplementary Fig. 10, ensuring the high colour routing performance and high colour imaging quality proven in the following measurements. Detailed information on the fabrication process is described in Supplementary Fig. 11 and in the Sample Fabrication section of Methods. Moreover, through the measurement of the light intensity distribution on the imaging plane, the routing function of the MBCR is further characterised using a homemade microscope system (Supplementary Fig. 13) with a white-light LED source (Supplementary Fig. 14). The measured light intensity distribution image at the imaging plane corresponding to a large-scale MBCR array when illuminated by a collimated white beam is shown in Fig. 3d. The RGB light is well routed to the target quadrants, showing good agreement between the experimental results and numerical simulations. The detailed light intensity distributions in the R, G and B channels under illumination with light of the three colours are depicted in Fig. 3e. Note that the light intensity distributions of the three primary colours have different patterns from the simulation results, which is due to the diffraction limit of the system. A small part of green light leaks into the quadrants belonging to the red and blue light, leading to a certain amount of colour crosstalk, as confirmed in the simulation results. This colour crosstalk will somewhat influence the quality of colour imaging and is difficult to suppress in the single-layer inverse-design metasurface. Based on this, we propose two workable solutions. First, we reduce the size of pillars to make more optimisation space available. After optimisation, we get some results that presents improved routing performance and similar efficiencies, as shown in Fig. r1. Second, metasurface hybridised with other devices might alleviate this problem. Here, we place a commercial Bayer CF below our MBCR to further eliminate crosstalk. The schematic diagram is shown in Supplementary Fig. 18.

Figure 3f presents the measured colour collection efficiencies of the MBCR in the visible spectrum region (400 nm–700 nm) (see Supplementary Note 3). Each colour curve corresponds to its colour channel. The black line represents the energy utilisation efficiency. The colour collection efficiencies reach peak values of 58, 59 and 49% at wavelengths of 640, 540 and 460 nm representing R, G and B light, respectively, which are much higher than those of typical commercial BCFs (see Supplementary Fig. 6). The energy utilisation efficiency is also plotted in Fig. 3f, providing an average value as high as 84%, which is twice as high as that of commercial BCFs. The low efficiency at the two ends of the working band is due to the low intensity of the illumination light source at these wavelengths (Supplementary Fig. 14), leading to a large error in the spectral measurement results.

**Comparison of imaging response between the metasurface-based colour router and colour filter.** To visualise the colour routing performance using the MBCR, we demonstrate colour imaging and compare the imaging results with those of commercial BCFs using a picture of a colourful apple (Fig. 4a). Figure 4a schematically shows the postprocessing procedure of colour imaging using the MBCR. A raw grey picture with a mosaic pattern is directly captured by the MBCR and

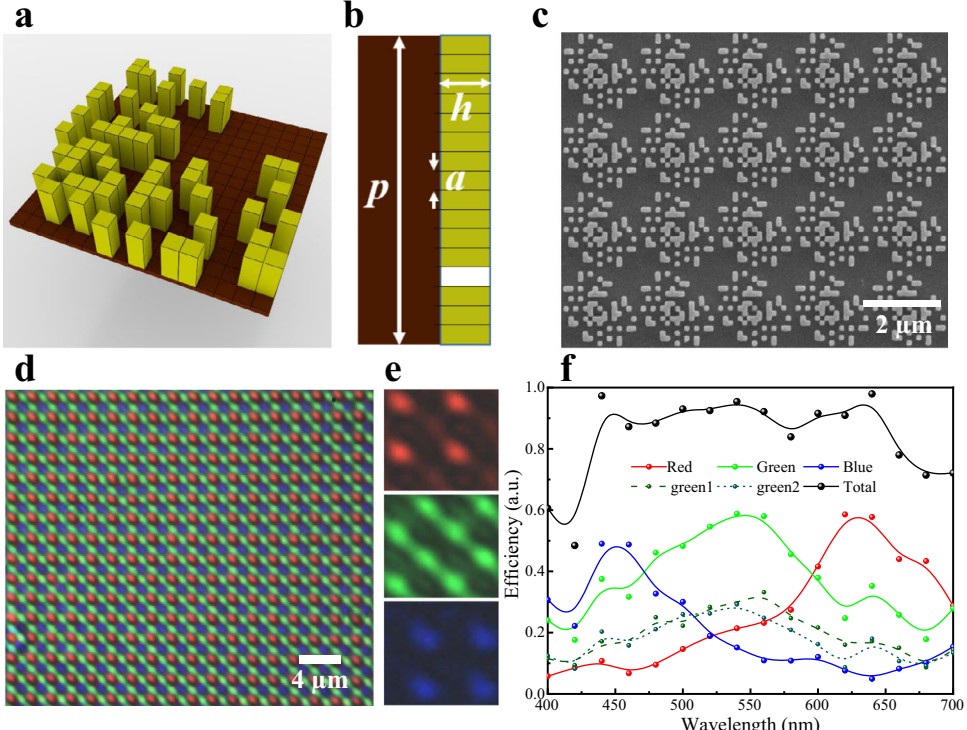

**Fig. 3 Experimental verification of colour routing response. a** Schematic of the optimised binary pattern of the MBCR with the unit cell size of 2 μm × 2 μm. The top layer $Si_3N_4$ patterns on a quartz glass substrate with a thickness of 500 μm. **b** Side view of the unit cell. Parameters of the MBCR are set as: $h = 600$ nm, $a = 125$ nm and $p = 2$ μm. **c** SEM image of a fabricated sample. **d** Measured image at the imaging plane of the MBCR array when illuminated by a collimated white beam. **e** Three colour channels of imaging plane under the illumination of three colour light. **f** The measured colour collection efficiencies of each channel in the visible region. Each colour curve corresponds to its colour channel. The black line represents the energy utilisation efficiency.

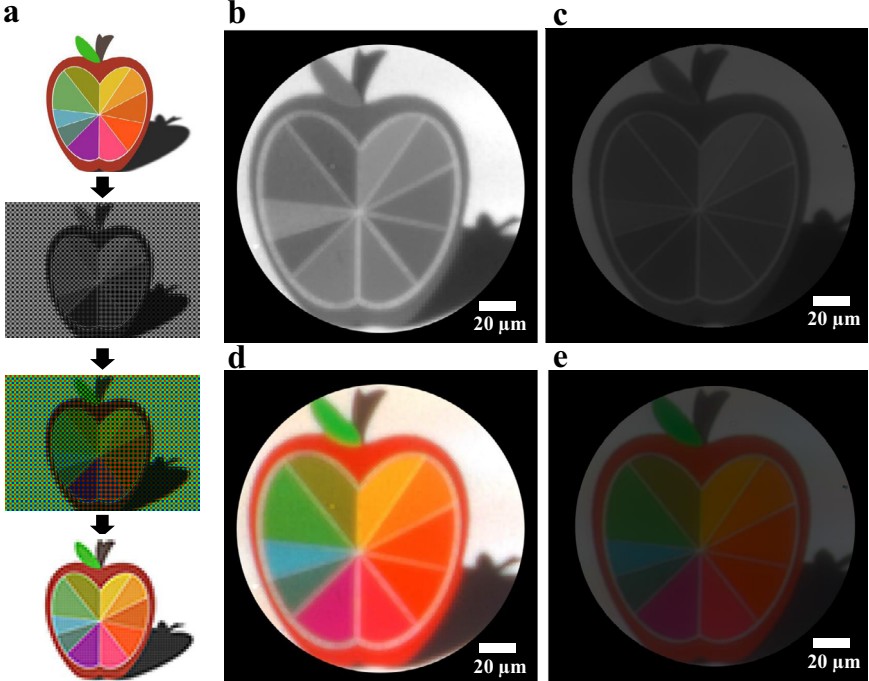

**Fig. 4 Comparison of imaging response between MBCR and BCFs. a** The postprocessing procedure of colour imaging. **b, c** Greyscale images were obtained directly by reconstructing the monochromatic images obtained using MBCR and BCFs respectively, for comparison of the intensity. **d, e** The reconstructed colour images of MBCR and BCFs combined with the spectral response, respectively.

monochromatic imaging sensor. Then, the spectral responses of the MBCR are imposed on each pixel to obtain the colour picture with a mosaic pattern using the conversion matrix method[18]. The conversion matrix directly converts the detected intensity of spectral values into three-channel RGB values. The final step is to demosaic the colour picture with a mosaic pattern. That is, according to the above RGGB information, the pixel interpolation algorithm (e.g., demosaicing interpolation) is carried out to obtain the RGB values of each RGGB unit cell. The colourful picture is ultimately reconstructed by transforming the RGGB pixel values to one three-channel pixel value, simulating the three-channel imaging in the actual colour imaging sensor.

For a comparison of the light intensity, two greyscale pictures (Fig. 4b, c) corresponding to the performance of the MBCR and commercial BCFs are reconstructed according to the directly captured greyscale images with mosaic patterns in Supplementary Fig. 20 using the above reconstruction process. The image captured using BCFs is obtained at the same position as that captured using the MBCR (see Supplementary Note 4). The image shown in Fig. 4b has an obviously higher brightness than that obtained by commercial BCFs (Fig. 4c). The enhancement factor of the imaging intensity is defined as the ratio between the two greyscale images, and the ratio for each pixel is calculated to be greater than 2, as shown in Supplementary Fig. 21. The average intensity enhancement factor is as high as 2.37, which is attributed to the colour routing using the metasurface mechanism. We process the raw image combined with the spectral response to achieve the colour image, and the resulting image is shown in Fig. 4d. Due to fabrication defects, some dark speckles can be seen in the captured image, leading to a decrease in the colour collection efficiency of the pixels, thus resulting in a colour cast of these pixels in the reconstructed image. The colour quality of the reconstructed image is nearly the same as that obtained with BCFs, which is shown in Fig. 4e. The imaging performance under off-normal incidence has also been experimentally studied, which gives a maximal numerical aperture (NA) value of 0.19 corresponding to an incident angle of 11°, as presented in Supplementary Fig. 22. Although the colour contrast slightly decreases and the image details are gradually lost, with the NA value increasing, the quality of colour imaging keeps quite good with the shapes and colours of the objects clearly distinguished, which is attributed to the high colour collection efficiencies and energy utilisation efficiency of MBCR. All these results demonstrate the robustness of our MBCR to oblique incidence.

## Discussion

In summary, we have demonstrated an MBCR with a pixel size conforming to that of a commercial imaging sensor. The MBCR can simultaneously sort and route more light to the imaging sensors than the conventional BCF. The average energy utilisation efficiency of the MBCR over the visible region is measured to be as high as 84%, which is twice as high as that of a commercial BCF. Moreover, colour imaging is realised using the MBCR sample in combination with a monochromatic imaging sensor. The resulting colour image presents higher brightness and the same colour quality compared with a commercial BCF. Our MBCR is quite suitable for direct integration with existing imaging sensors by using common semiconductor processing technology. We anticipate that the MBCR will enable the creation of high-efficiency and miniaturised colour imaging sensors for various emerging applications.

## Methods

**Design and simulation**. In this paper, relying on the genetic algorithm toolbox in Matlab, and using the commercial FDTD software (Lumerical FDTD solutions) with joint function, a simulation system combining Matlab and FDTD is

established. We divide the unit cell into $16 \times 16$ squares to optimise the binary arrangement of silicon nitride ($Si_3N_4$) nanopillars. The pixel pitch is 1 μm and the pitch of the unit cell of the metasurface is 2 μm and the thickness of the whole sample is 500.6 μm (600 nm silicon nitride pillars and 500 μm quartz substrate). The smallest silicon nitride pillar dimension can be arbitrarily reduced to the size of the computing grid in the simulation calculation, such as 10 nm, but it will require a large number of computer memory and computing time. However, if the aspect ratio of silicon nitride nanostructures is higher than 5:1, some isolated pillars of metasurface may collapse and the expected metasurface pattern will not be obtained. Thus, considering the common limitation of nano-fabrication conditions (i.e., the best aspect ratio of silicon nitride nanostructures is 5:1), we set the minimum side length of the pillar to 125 nm and the height of the pillar to 600 nm. The refractive index of $Si_3N_4$ is shown in Supplementary Fig. 1. First, we randomly generate N two-dimensional code structures, which can be mathematically represented by a 0-1 code sequence. Then the individuals of the population in the genetic algorithm, that is, the binary matrix containing the information of metasurface structure arrangement, are transferred to the FDTD simulation software. We build the two-dimensional code structural models in FDTD, set the periodic boundary conditions in ±x and ±y directions and PML conditions in ±z directions, and select the simulation frequency band of the plane wave incident light source. Set four transmittance monitors on the imaging plane, which corresponds to four quadrants of an area of $2 \times 2$ μm² on the optical axis. The transmittance $T_R$, $T_{G1}$, $T_{G2}$ and $T_B$ of the four monitors are calculated by solving the Maxwell equations in FDTD. According to the obtained $T_R$, $T_{G1}$, $T_{G2}$ and $T_B$, the fitness function value $F$ is calculated. Afterwards, the calculated corresponding fitness function $F$ is assigned to the corresponding parameter variables in Matlab. We judge whether the fitness function values of N structures reach the optimal design goal. And if so, terminate the optimisation process; if not, generate N probability values based on the evaluation function values of these N structures. The principle of natural selection is that the higher the fitness function value is, the higher the probability of being selected and is defined as

$$P_i = \frac{F_i^2}{\sum F_i^2}. \tag{2}$$

The newly selected N structures are mutated and crossed according to a certain ratio to obtain the next generation of N two-dimensional code structures. The above steps are repeated in the simulation system until reaching the termination condition. The termination condition is set to be that the improvement of the global fitness function value is less than 0.000001 within 200 generations (2000 solutions) of iterations.

**Sample fabrication**. A quartz glass substrate with a 500-μm thickness is prepared using a cleaning process. A 600-nm-thick silicon nitride film is deposited over the cleaned substrate by plasma-enhanced chemical vapour deposition (PECVD). Conductive positron beam photoresist (CSAR-6200, a positive tone electron-beam resist) with a thickness of 120 nm is then spin-coated. The optimised metasurface pattern is defined in the CSAR-6200 resist by electron beam lithography (EBL, Elionix, ELS-F125-G8). After the development of the resist, a layer of chromium (Cr) with the pattern array is evaporated onto the sample by electron beam evaporation (EBE). After performing lift-off, a Cr hard mask is left on the $Si_3N_4$ layer for subsequent etching. The sample is etched using an inductively coupled plasma and reactive ion etching etcher (ICP-RIE) with a mixture of $O_2$ and $CHF_3$. Eventually, the rest Cr hard mask is removed using a corrosion solution called ammonium cerium nitrate. The flow chart of fabrication is shown in Supplementary Fig. 11.

## Data availability

Relevant data supporting the key findings of this study are available within the article and the Supplementary Information file. All raw data generated during the current study are available from the corresponding author upon reasonable request.

## Code availability

The code used for analyses and figures is available from the corresponding author upon reasonable request.

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

## Acknowledgements

The authors acknowledge financial support from the National Programme on Key Basic Research Project of China (2017YFA0303700) and the National Natural Science Foundation of China (no. 11621091, 11822406, 11834007, 11774164 and 11774162).

## Author contributions

X.Z. and Y.Z. contributed equally to this research. Y.Z. and S.W. conceived the original idea. Y.Z. and X.Z. developed the design and optimisation of the metasurface. X.Z., S.W., R.L. and G.G. performed the fabrication of the metasurface and developed the measurement and data analysis of the experiment. S.W., Y.Z., Z.W. and S.Z. supervised the research. All the authors discussed the contents and prepared the manuscript.

## Competing interests

The authors declare no competing interests.
