## [Peer Review File · Nature Communications]

Pixel-level Bayer-type color router based on metasurfacesREVIEWER COMMENTS

Reviewer #1 (Remarks to the Author):

Color filter arrays are ubiquitous components in consumer optics and digital cameras.

The Bayer design achieves color discrimination by absorption based color filtering which affects the overall image capture efficiency. Metasurfaces are highly promising replacements for traditional pigment based CFAs.

In their contribution, the authors demonstrate a metasurface based alternative to traditional CFAs. The noteworthy results are the more than 2x image capture efficiency. The strongest point of the manuscript is the demonstrated improvement over traditional Bayer with a design that is CMOS process compatible as well.

There are, in my opinion, some deficiencies which necessitate a thorough revision and I can recommend publication once the revision is satisfactory.

1. The literature survey is not comprehensive. I suggest to reference non optical techniques that solve the Bayer inefficiency problem including stacked color dependent photodiode. I also suggest a comprehensive look at metasurface transmission filters and comparison of transmission efficiencies.

2. Bayer filters of rrgb allocate 25% of a pixel to each color. Does this not imply that Rand B pixels are limited to 1/4 not 1/3 as written in the introduction.

3. The pixel pitches, the critical dimension and other details are not clearly evident. What led to the smallest silicon nitride pillar dimension. How does the performance drop if we are limited by the smallest feature size? Discuss in the context of current photo lithography resolution constraints.

4. The performance of routing is not assessed by using color matching functions. Instead, it approximately divides the visible spectrum. I think a more exact quantitative study in terms of CMF is needed.

5. The behavior of the metasurface for inclined illumination is not considered. If the behavior is highly incident angle dependent, that will limit the imaging to shallow angles only. This limitation needs to be addressed, at least, in simulation.

6. There are several English language issues with the submission that detracts the reader. I recommend a thorough and professional proof reading and editing.

Reviewer #2 (Remarks to the Author):

In this manuscript, the authors demonstrate a metasurface-based bayer color filter designed using an inverse-design approach. The authors extend this to a fairly large area, 200 microns x 200 microns to show full color imaging with this approach. I think the approach and results are fairly interesting, but I have several concerns about novelty as well as the results that preclude me from recommending in favor of publication at this time.

- Color filters, including pixel scale color splitters, using meta surfaces have been explored in depth over the last five years. As one example, this paper describes an effective scheme:

<https://pubs.acs.org/doi/abs/10.1021/acsp Photonics.9b00042>

Here is another: <https://www.spiedigitallibrary.org/conference-proceedings-of-spie/10671/106711F/Color-filter-arrays-based-on-dielectric-metasurface-elements/10.1117/12.2307155.short?SSO=1>

While the authors do reference many important metasurface imaging-based papers, I would suggest they specifically contrast with the literature on metasurface-based color filters, which seems rather extensive.

- The authors state in the Supplementary Information that the genetic algorithm continues evolving the design till the 'target' is reached. But clearly the color filter performance is not optimal so it is not clear to me that a 'target' was in fact reach. How was the decision to terminate the optimization made? Why did the authors use this method and not more standard ones in photonics literature such as topology/ adjoint optimization?

- A potential issue with the style of design yielded by the inverse design algorithm is its highly 'pixelated' nature. How close are the fabricated structures to the inverse-design metasurface? Relatedly, how does

the color filter performance compare to the Numerical simulations? There is no simulation data as far as I can see.

- The performance of the green filtering seems quite poor to me. While blue and red seem fine, if I look at Fig. S3, the green power distribution seems almost random. This can be seen in Fig. 3F. I am surprised the authors state that there is little crosstalk issue when the power efficiency at say 500 nm is 50% for the green but is 0.4 and 650 and 700nm. This suggests to me the design is not nearly optimal for green - in fact the design seems to have optimized for good blue and red discrimination, but for green is fairly randomized.

- What is the angular response of this metasurface filter? Presumably all the measurements have been at exactly normal incidence. But a traditional BCF does have reasonable angular acceptance range. I suspect this metasurface would not perform well off-normal. Either simulations or experimental data would be important to have here to understand the limitations of the metasurface.

- Finally, there are quite a few grammatical errors and awkward phrasings in the manuscript. It could benefit from a closer reading/ editing for this.

Reviewer #3 (Remarks to the Author):

Instead of traditional filtering, the authors introduced the method of routing R, G, B colors into different regions, therefore, increases efficiency. The structure is only single-layer, and is designed using the GA optimization. Experiments demonstrated a 84% color collection efficiency over visible band. The designed structures can be directly applied to current Bayer-type image sensor to solve the low efficiency problem. The manuscript can be accepted with revision if the following issues are well-addressed.

At line 102, the author mentioned the genetic algorithm is used for optimization. There are also other optimization methods, e.g., particle swarm, differential evolution, Bayesian optimization. Please explain why authors specifically choose the genetic algorithm.

The genetic algorithm and the design process are not well discussed. Some information should be implemented or added:

(a) Fig. 2b seems too simplified. It is better to give more details on how GA works.

(b) The objective of the fitness function at line 124 should be discussed. At line 96-97, the authors mention “the routing property of the MBCR is adopted to replace both the XXXXX”. How this routing is reflected in the fitness function? How routing is adopted to replace XXX? Please add some discussions.

(c) Some hyperparameters as well as the computer resources of GA algorithms can be provided for referencing. Some intermediate results during GA algorithms can be provided, e.g., the fitness function vs. epoch during optimization, several examples of generated structures during optimization.

(d) More illustrations of why this structure can give the routing ability is needed, which can help researchers to understand the physics. Some pointing vector distributions are given in In Fig. 2c-e. However, this is not sufficient. It would be better to also provide the power flow in the z direction/ in 3d, and the electrical field at the designed structures.

(e) In Fig. 2d, there are some crosstalk for the green light. I wonder if this is because of the fitness function, where you want to distribute green light to two different regions. Will the crosstalk decrease if you only let the green light to route into one region? In terms of real application, there is no need to exactly follow the light distribution of Bayer-type image sensor, as long as they are compatible.

Some other minor comments:

1. At line 47-49, the reason why using metallic plasmonic structures or photonic crystal structures can improve the transmission can be briefly discussed.

2. From line 53-79, many discussions are not necessary. Please delete some redundant information to make it concise. Similar issues can be found at line 97-101, where the inverse design was already mentioned in the introduction.

3. In Fig. 2c-e, the calculated Poynting vector distributions are provided to visualize the power flow density distributions. The blue one is hard to see in the dark background, is it better to view in white

background? Also, what is the light distribution slightly outside the color pixel? Is there any color crosstalk at nearby effective color pixel?

4. From line 115-117, I am confused what the authors refer to for Fig. 2(a). Is this the structure at the image plane, or this is the metasurface? Please be clear.

5. At line 170, the author mentioned "The RGB lights are well routed and focused into their target quadrants". It would be better to describe where the focusing happens.

6. At line 165, and also 172-173, the author mention the field distribution is given in Fig. 3e. Field distribution usually contains both amplitude and phase, here is only the intensity of E field.

7. The method of reconstruction of colorful image at line 195-198 is not clearly stated.

8. Fig. S4 and S5 are not compatible, there is no total efficiency in S5 for comparison.

9. Fig. S11, some figures are too dark too be seen. Maybe improving the brightness or contrast will help?

To Reviewer 1:

Color filter arrays are ubiquitous components in consumer optics and digital cameras. The Bayer design achieves color discrimination by absorption based color filtering which affects the overall image capture efficiency. Metasurfaces are highly promising replacements for traditional pigment based CFAs. In their contribution, the authors demonstrate a metasurface based alternative to traditional CFAs. The noteworthy results are the more than 2x image capture efficiency. The strongest point of the manuscript is the demonstrated improvement over traditional Bayer with a design that is CMOS process compatible as well. There are, in my opinion, some deficiencies which necessitate a thorough revision and I can recommend publication once the revision is satisfactory.

Our reply: We thank the reviewer for the deep understanding of our work and the positive comments on it. We have answered all the concerns point-to-point, and revised the manuscript accordingly.

1. The literature survey is not comprehensive. I suggest to reference non optical techniques that solve the Bayer inefficiency problem including stacked color dependent photodiode. I also suggest a comprehensive look at metasurface transmission filters and comparison of transmission efficiencies.

Our reply: Thanks for the helpful suggestion. Citations of literatures on stacked color dependent photodiodes have been added in the **Introduction** of our revised manuscript [a-c] (*Ref. 10-12 in the main text*). Stacked color dependent photodiodes are new types of color image sensors based on organic photoconductive films to improve light utilization efficiency, but the large pixel size and the synthesis of complex organic photoelectric conversion modules hinder their application in the color cameras. Some improved color filter combinations, such as RYYB and RGBW found in the market nowadays (see <https://inf.news/en/digital/23a60abb65e44270f65b6b54d3af3b9.html>), have been introduced to increase the light utilization efficiency, but it simultaneously brings another problem of color cast, which requires complex post-processing to obtain the correct color.

According to the reviewer's suggestion, more works about the transmission color filters [e-g] have also been added in the references (*Ref. 18, 9, and 19 in the main text*). Moreover, to show the benefit of our color router, a comprehensive comparison of the transmission efficiency of these color routers/filters is made, which is shown in the following table. Beside of the efficiency, the device

configuration (Bayer-type or not, router or filter), pixel size (pixel-level or not), and realization form (theory or experiment, imaging or not) have also been compared between various color filters/routers, as shown in the below table. Since the efficiencies of color filters are limited by the filtering mechanism, the color router configuration is a better choice, which does not have such limitation. Among different color routers, the Bayer-type color router is preferred and is suitable for the commonly used RGGB arrangement of CMOS sensor. Although the color routers based on the forward-designed metasurface can achieve color sorting and focusing at specific wavelengths, the blue light presents a low efficiency, which is not conducive to practical applications [g]. Numerical simulations of the inverse-designed 3D irregular nanostructures via topology optimization show good routing effects, but these 3D nanostructures are difficult to realize via the state of art nanofabrication technologies [h, i]. Take all these issues into account, our color router based on single-layer metasurface can be considered as the optimal choice, which owns high energy utilization efficiency, simple sample preparation and high-quality imaging for pixel-level pitches ($\sim 1\mu\text{m}$).

Table. Comparison between reported transmission color imaging configurations based on nanostructures.

PUBLICATION	TITLE	BAYER-TYPE OR NOT	ROUTER OR FILTER	EFFICIENCY#	PIXEL-LEVEL	THEORY OR EXPERIMENT	IMAGING OR NOT
Nature Communications, 2010, 1, 1-5	Plasmonic nanoresonators for high-resolution colour filtering and spectral imaging [6]	NO	FILTER	(45%,65%,55%)/3	NO	EXPERIMENT	YES
NanoLetters, 2012, 12, 4349-4354	Plasmonic Color Filters for CMOS Image Sensor Applications [7]	NO	FILTER	(40-50%)/3	YES	EXPERIMENT	NO
ACS Nano, 2013, 7, 10038-10047	Color imaging via nearest neighbor hole coupling in plasmonic color filters integrated onto a complementary metal-oxide semiconductor image sensor [8]	YES	FILTER	50%/3	NO	EXPERIMENT	YES
Scientific Reports, 2015, 5, 8467	Omnidirectional color filters capitalizing on a nanoresonator of Ag-TiO ₂ -Ag integrated with a phase compensating dielectric overlay [d]	NO	FILTER	(40%,65%,65%)/3	NO	EXPERIMENT	NO
NanoLetters, 2017, 17, 3159-3164	Visible wavelength color filters using dielectric subwavelength gratings for backside-illuminated CMOS image sensor technologies [4]	YES	FILTER	(60-80%)/3	YES	EXPERIMENT	NO
Metamaterials XI, 2018, 10671, 106711F	Color filter arrays based on dielectric metasurface elements [b]	NO	FILTER	(70-90%)/3	NO	THEORY	NO
ACS photonics, 2019, 6, 1442-1450	Submicrometer Nanostructure-Based RGB Filters for CMOS Image Sensors [a]	YES	FILTER	(60%-80%)/3	YES	EXPERIMENT	YES
Nature Photonics, 2013, 7, 240-246	Efficient colour splitters for high-pixel-density image sensors [17]	NO	ROUTER	—	NO	EXPERIMENT	YES
Optica, 2015, 2, 933-939	Ultra-high-sensitivity color imaging via a transparent diffractive-filter array and computational optics [20]	NO	ROUTER	—	NO	EXPERIMENT	YES
NanoLetters, 2017, 17, 6345-6352	GaN metalens for pixel-level full-color routing at visible light [16]	YES	ROUTER	15.9%, 65.42%, 38.33%	NO	EXPERIMENT	NO

Optica, 2019, 6, 1367-1373	Integrated nanophotonic wavelength router based on an intelligent algorithm [21]	NO	ROUTER	50%-70%	NO	EXPERIMENT	NO
ACS Photonics, 2019, 6, 1442-1450	High-sensitivity color imaging using pixel-scale color splitters based on dielectric metasurfaces[18]	NO	ROUTER	40–50%	NO	EXPERIMENT	YES
Optica, 2020, 7, 280-283	Multifunctional volumetric meta-optics for color and polarization image sensors [f]	YES	ROUTER	~57%	YES	THEORY	NO
Nanoscale, 2021, 13, 13024-13039	Full-color nanorouter for high-resolution imaging	NO	ROUTER	50%,50%,50%	YES	THEORY	NO
Optica, 2021, 8, 1596-1604	Full-color-sorting metalenses for high-sensitivity image sensors [e]	YES	ROUTER	48%,48%,12%	YES	EXPERIMENT	YES
	OUR WORK	YES	ROUTER	58%,59%,49%	YES	EXPERIMENT	YES

In the table, those color router/filter cases with efficiencies higher than the efficiencies of an ideal color filter, i.e., 25%, 50%, and 25% at Red, Green, and Blue light are marked with red.

- [a] Sakai, T. et al. Color image sensor with organic photoconductive films. *IEEE International Electron Devices Meeting (IEDM)* 30.33. 31-30.33. 34 (2015).
- [b] Togashi, H. et al. Three-layer stacked color image sensor with 2.0- μm pixel size using organic photoconductive film. *IEEE International Electron Devices Meeting (IEDM)* 16.16. 11-16.16. 14 (2019).
- [c] Lim, S.J. et al. Organic-on-silicon complementary metal-oxide-semiconductor colour image sensors. *Sci Rep* 5, 7708 (2015).
- [d] Park, C., Shrestha, V., Lee, S., et al. Omnidirectional color filters capitalizing on a nano-resonator of Ag-TiO₂-Ag integrated with a phase compensating dielectric overlay. *Sci Rep* 5, 1-8 2015.
- [e] Miyata, M., Nakajima, M. & Hashimoto, T. High-Sensitivity Color Imaging Using Pixel-Scale Color Splitters Based on Dielectric Metasurfaces. *ACS Photonics* 6, 1442-1450 (2019).
- [f] Berzins, J. et al. Color filter arrays based on dielectric metasurface elements. *International Society for Optics and Photonics (Metamaterials XI)* 10671, 106711F (2018).
- [g] Miyata, M., Nemoto, N., Shikama, K., Kobayashi, F. & Hashimoto, T. Full-color-sorting metalenses for high-sensitivity image sensors. *Optica* 8, 1596-1604 (2021).
- [h] Camayd-Muñoz, P., Ballew, C., Roberts, G. & Faraon, A. Multifunctional volumetric meta-optics for color and polarization image sensors. *Optica* 7, 280-283 (2020).
- [i] Johlin, E. Nanophotonic color splitters for high-efficiency imaging. *iScience* 24, 102268

(2021).

2. Bayer filters of rrgb allocate 25% of a pixel to each color. Does this not imply that R and B pixels are limited to 1/4 not 1/3 as written in the introduction.

Our reply: Thanks for raising this point. The energy utilization efficiency ($\sim 1/3$ ideally) mentioned in the main text refers to the ratio of light intensity detected by a single color pixel to that of the total incident light, when a white light with the identical intensity at any wavelength in visible region is incident. Assuming that RGB light respectively covers 1/3 of the total incident energy, the energy utilization efficiency can be expressed as $\frac{1}{3}T_R + \frac{1}{3}T_G + \frac{1}{3}T_B = \frac{1}{3}$, where the color collection efficiencies of the RGB light of Bayer color filter are ideally $T_R = 1/4$, $T_G = 1/2$, and $T_B = 1/4$, respectively. We have modified the corresponding description in the **Introduction** of the revised manuscript as follows:

The ideal energy utilization efficiency defined as the ratio of the light intensity detected by a single color pixel to that of the total incident light is limited to 1/3, for an incident white light with the identical intensity in the visible region and the actual loss of the CFs themselves are not considered.

3. The pixel pitches, the critical dimension and other details are not clearly evident. What led to the smallest silicon nitride pillar dimension. How does the performance drop if we are limited by the smallest feature size? Discuss in the context of current photo lithography resolution constraints.

Our reply: In this work, the pixel pitch and the size of unit cell of metasurface are respectively chosen as $1\ \mu\text{m}$ and $2 \times 2\ \mu\text{m}^2$, which is consistent with the size of commercial imaging sensors. The thickness of the metasurface is fixed to 600 nm for the optimization of the transmission and working efficiencies. Considering the feature size of common nano-fabrication technology is about 10 nm and the aspect ratio of silicon nitride nanostructures is about 5:1, we choose the dimension of the pillars to be $125\text{nm} \times 125\text{nm} \times 600\text{nm}$. In fact, the side length of silicon nitride pillar can be further reduced in simulation calculation to present a slight improvement in color routing effect, such as the size of pillar being $100\text{nm} \times 100\text{nm} \times 600\text{nm}$ (as shown in **Fig. r1**). However, when the side length is smaller than 125 nm, some isolated pillars of metasurface may collapse, and the

expected metasurface pattern will not be obtained. In experiment, we have tried to fabricate such smaller nano-pillars with a higher aspect ratio of 7.5:1 with a side length of 80 nm. The fabricated samples always have some defects as is shown in **Fig. r2**. Therefore, in our work, the dimension of the nano-pillar is fixed to $125\text{nm} \times 125\text{nm} \times 600\text{nm}$, which satisfies the optimization of the color routing functions.

Fig. r1. The color router structure designed with pillar size of $100\text{nm} \times 100\text{nm} \times 600\text{nm}$. The solid lines in the spectral efficiency diagram represent the efficiencies of the color router (Subscript 0) reported in our work, while the dotted lines represent the efficiencies of the color router (Subscript 1) designed with a smaller pillar size.

Fig. r2. SEM diagram of nano-pillars with aspect ratio of 7.5:1.

4. The performance of routing is not assessed by using color matching functions. Instead, it approximately divides the visible spectrum. I think a more exact quantitative study in terms of CMF is needed.

Our reply: Thanks for the valuable suggestion. We have calculated the tri-stimulus values (X, Y, Z) for a light source with spectral distribution $T(\lambda)$, according to the following equations,

$$X = \int_{400}^{700} \bar{x}(\lambda)T(\lambda)d\lambda$$
$$Y = \int_{400}^{700} \bar{y}(\lambda)T(\lambda)d\lambda$$
$$Z = \int_{400}^{700} \bar{z}(\lambda)T(\lambda)d\lambda$$

where $\bar{x}(\lambda)$, $\bar{y}(\lambda)$, $\bar{z}(\lambda)$ are the CIE XYZ color matching functions. X, Y and Z are normalized

to obtain the chromaticity coordinates (x, y, z), as follows:

$$x = X / (X + Y + Z)$$
$$y = Y / (X + Y + Z) .$$
$$z = Z / (X + Y + Z)$$

Finally, the three chromaticity coordinates (x_R, y_R, z_R) and (x_G, y_G, z_G) and (x_B, y_B, z_B) in the 1931 CIE chromaticity diagram are calculated and shown in **Fig. s17**, using the spectral real lines of three-color pixels shown in **Fig. 3f** in the paper (here it is provided in **Fig. s18b**). In the diagram, colors obtained by MBCR (R1, G1, B1) and common CFs (R2, G2, B2) are represented in the color space, respectively. The color contrast of our RGB colors is lower than that of CFs, which can be attributed to the crosstalk between different colors. In fact, to solve this problem, we can insert a commercial Bayer CF below our MBCR to further eliminate this crosstalk. The schematic diagram is shown in **Fig. s18a**. By multiplying the measured spectra (see **Fig. s18b**) with the transmittance of commercial BCFs (see **Fig. s6**), we can calculate the color collection efficiencies of this new design, which can both present the small color crosstalk of the CF and also keep the high efficiencies of MBCR, as shown in **Fig. s18c**. The chromaticity coordinates of this crosstalk cancellation scheme are marked with R3, G3, B3 in **Fig. s17**, with a better color contrast than that of the CFs. Since we

only focus on a better single layer CR with much higher efficiencies than the commercial BCF, such compound design will not be investigated in this work.

The discussions on this issue have been added to the Section 3 (the third paragraph) of the Supplementary Information.

Fig. s17. Characterization of colors obtained by MBCR (R1, G1, B1), commercial CFs (R2, G2, B2) and the crosstalk cancellation scheme by combining our CR and BCF (R3, G3, B3) in the 1931 CIE chromaticity diagram.

Fig. s18. (a) Schematic of the crosstalk cancellation scheme that combine MBCR and CFs together. (b) (c) The measured color collection efficiencies of each channel in the visible region. Each color curve corresponds to its color channel. The black line represents the energy utilization efficiency. (b) represents the measured efficiencies by MBCR, while (c) refers to efficiencies obtained by the crosstalk cancellation scheme.

5. The behavior of the metasurface for inclined illumination is not considered. If the behavior is highly incident angle dependent, that will limit the imaging to shallow angles only. This limitation needs to be addressed, at least, in simulation.

Our reply: Thanks for pointing out this important issue. The angular response for oblique incidence has been numerically simulated, which is shown in **Fig. s9**. It is seen that with the increasing of the incident angle, the peak values of all color collection efficiencies decrease gradually. Numerical simulation shows a maximum acceptable incident angle of about 30° , with the color routing function remaining. Under larger incident angles (40° and 50°), serious crosstalk occurs between RGB pixels, and the color routing function cannot be realized. The imaging performance under off-normal incidence has also been experimentally studied, which gives a maximal numerical aperture (NA) value of 0.19 corresponding to an incident angle of 11° , as shown in **Fig. s22**. Although the color contrast slightly decreases and the image details are gradually lost, with the NA value increasing, the quality of color imaging keeps good with the shapes and colors of the objects clearly

distinguished, which is attributed to the high color collection efficiencies and energy utilization efficiency of MBCR. All these results demonstrate the robustness of our MBCR to oblique incidence.

The above discussions have been added to the Results section (the fourth paragraph and eighth paragraph) of the manuscript.

Fig. s9. Transmittance curves of our MBCR when considering the angle of incidence up to 50° .

Fig. s22. Influence of numerical aperture (NA) on imaging performance. (a) Images taken with different numerical aperture (NA) being 0.02, 0.05, 0.09, 0.14, and 0.19, from left to right. (b) Image reconstruction of the pictures in (a).

6. There are several English language issues with the submission that detracts the reader. I recommend a thorough and professional proof reading and editing.

Our reply: The language mistakes have been corrected. The whole manuscript has also been checked further.

To Reviewer 2:

In this manuscript, the authors demonstrate a metasurface-based bayer color filter designed using an inverse-design approach. The authors extend this to a fairly large area, 200 microns x 200 microns to show full color imaging with this approach. I think the approach and results are fairly interesting, but I have several concerns about novelty as well as the results that preclude me from recommending in favor of publication at this time.

Our reply: Thanks a lot for the positive comment on this work! We have addressed all the concerns point-to-point and revised the manuscript accordingly, which can further clarify the novelty and significance of this work. We hope that the new version can be up to your requirement for publication.

1. Color filters, including pixel scale color splitters, using metasurfaces have been explored in depth over the last five years. As one example, this paper describes an effective scheme: <https://pubs.acs.org/doi/abs/10.1021/acsphotonics.9b00042>

Here is another: <https://www.spiedigitallibrary.org/conference-proceedings-of-spie/10671/106711F/Color-filter-arrays-based-on-dielectric-metasurface-elements/10.1117/12.2307155.short?SSO=1>

While the authors do reference many important metasurface imaging-based papers, I would suggest they specifically contrast with the literature on metasurface-based color filters, which seems rather extensive.

Our reply: Thanks for the helpful suggestion. We have added the papers mentioned above [*a*, *b*] and two more related works [*c*, *e*] to the references of the manuscript (*Ref. 10-12, 18 in the manuscript*) to form a comprehensive look at the metasurface-based color filters/routers. The device configuration (Bayer type or not, router or filter), efficiency, pixel size (pixel-level or not), and realization form (theory or experiment, imaging or not) have been compared between various color filters/routers, shown in the below table. Since the efficiencies of color filters are limited by the filtering mechanism, the color router configuration is a better choice, which does not have such limitation. Among different color routers, the Bayer-type color router is preferred that is suitable for the commonly used RGGGB arrangement of CMOS sensor. Although the color routers based on the forward-design can achieve good color sorting and focusing at specific wavelengths, one of the colors has a low efficiency, which is not conducive to practical applications [*e*] (*Ref. 18 in the*

manuscript). In this work, we employ an inverse-design approach to achieve as high efficiencies as possible to overcome the intrinsic limit of the commercial Bayer color filter, with the collection efficiencies of RGB light all larger than ideal Bayer color filters (1/4, 1/2, and 1/4 for RGB light). In fact, the 3D structures or multi-layer structures designed by topology optimization theoretically show good color routing function [f , g] (*Ref. 9, 19 in the manuscript*), however, they are quite difficult to be realized due to the limitation of the state of art of 3D nano-fabrication, especially for the high frequencies, such as near-infrared or visible range. In summary, our color router based on single-layer metasurface can be considered as the optimal choice, which owns high efficiencies, suitable device configurations, simple sample preparation and high-quality imaging for pixel-level pitches ($\sim 1 \mu\text{m}$).

Table. Comparison between reported transmission color imaging configurations based on nanostructures.

PUBLICATION	TITLE	BAYER-TYPE OR NOT	ROUTER OR FILTER	EFFICIENCY#	PIXEL-LEVEL	THEORY OR EXPERIMENT	IMAGING OR NOT
Nature Communications, 2010, 1, 1-5	Plasmonic nanoresonators for high-resolution colour filtering and spectral imaging [6]	NO	FILTER	(45%,65%,55%)/3	NO	EXPERIMENT	YES
NanoLetters, 2012, 12, 4349-4354	Plasmonic Color Filters for CMOS Image Sensor Applications [7]	NO	FILTER	(40-50%)/3	YES	EXPERIMENT	NO
ACS Nano, 2013, 7, 10038-10047	Color imaging via nearest neighbor hole coupling in plasmonic color filters integrated onto a complementary metal-oxide semiconductor image sensor [8]	YES	FILTER	50%/3	NO	EXPERIMENT	YES
Scientific Reports, 2015, 5, 8467	Omnidirectional color filters capitalizing on a nanoresonator of Ag-TiO ₂ -Ag integrated with a phase compensating dielectric overlay [d]	NO	FILTER	(40%,65%,65%)/3	NO	EXPERIMENT	NO
NanoLetters, 2017, 17, 3159-3164	Visible wavelength color filters using dielectric subwavelength gratings for backside-illuminated CMOS image sensor technologies [4]	YES	FILTER	(60-80%)/3	YES	EXPERIMENT	NO
Metamaterials XI, 2018, 10671, 106711F	Color filter arrays based on dielectric metasurface elements [b]	NO	FILTER	(70-90%)/3	NO	THEORY	NO
ACS photonics, 2019, 6, 1442-1450	Submicrometer Nanostructure-Based RGB Filters for CMOS Image Sensors [a]	YES	FILTER	(60%-80%)/3	YES	EXPERIMENT	YES
Nature Photonics, 2013, 7, 240-246	Efficient colour splitters for high-pixel-density image sensors [17]	NO	ROUTER	—	NO	EXPERIMENT	YES
Optica, 2015, 2, 933-939	Ultra-high-sensitivity color imaging via a transparent diffractive-filter array and computational optics [20]	NO	ROUTER	—	NO	EXPERIMENT	YES
NanoLetters, 2017, 17, 6345-6352	GaN metalens for pixel-level full-color routing at visible light [16]	YES	ROUTER	15.9%, 65.42%, 38.33%	NO	EXPERIMENT	NO
Optica, 2019, 6, 1367-1373	Integrated nanophotonic wavelength router based on an intelligent algorithm [21]	NO	ROUTER	50%-70%	NO	EXPERIMENT	NO
ACS Photonics, 2019, 6, 1442-1450	High-sensitivity color imaging using pixel-scale color splitters based on dielectric metasurfaces[18]	NO	ROUTER	40–50%	NO	EXPERIMENT	YES

Optica, 2020, 7, 280-283	Multifunctional volumetric meta-optics for color and polarization image sensors [f]	YES	ROUTER	~57%	YES	THEORY	NO
Nanoscale, 2021, 13, 13024-13039	Full-color nanorouter for high-resolution imaging	NO	ROUTER	50%,50%,50%	YES	THEORY	NO
Optica, 2021, 8, 1596-1604	Full-color-sorting metalenses for high-sensitivity image sensors [e]	YES	ROUTER	48%,48%,12%	YES	EXPERIMENT	YES
	OUR WORK	YES	ROUTER	58%,59%,49%	YES	EXPERIMENT	YES

In the table, those color router/filter cases with efficiencies higher than the efficiencies of an ideal color filter, i.e., 25%, 50%, and 25% at Red, Green, and Blue light are marked with red.

- [a] Miyata, M., Nakajima, M. & Hashimoto, T. High-Sensitivity Color Imaging Using Pixel-Scale Color Splitters Based on Dielectric Metasurfaces. *ACS Photonics* 6, 1442-1450 (2019).
- [b] Berzins, J. et al. Color filter arrays based on dielectric metasurface elements. *International Society for Optics and Photonics (Metamaterials XI)* 10671, 106711F (2018).
- [c] Lim, S.J. et al. Organic-on-silicon complementary metal-oxide-semiconductor colour image sensors. *Sci Rep* 5, 7708 (2015).
- [d] Park, C., Shrestha, V., Lee, S., et al. Omnidirectional color filters capitalizing on a nano-resonator of Ag-TiO₂-Ag integrated with a phase compensating dielectric overlay. *Sci Rep* 5, 1-8 2015.
- [e] Miyata, M., Nemoto, N., Shikama, K., Kobayashi, F. & Hashimoto, T. Full-color-sorting metalenses for high-sensitivity image sensors. *Optica* 8, 1596-1604 (2021).
- [f] Camayd-Muñoz, P., Ballew, C., Roberts, G. & Faraon, A. Multifunctional volumetric meta-optics for color and polarization image sensors. *Optica* 7, 280-283 (2020).
- [g] Johlin, E. Nanophotonic color splitters for high-efficiency imaging. *iScience* 24, 102268 (2021).

2. The authors state in the Supplementary Information that the genetic algorithm continues evolving the design till the 'target' is reached. But clearly the color filter performance is not optimal so it is not clear to me that a 'target' was in fact reach. How was the decision to terminate the optimization made? Why did the authors use this method and not more standard ones in photonics literature such as topology/ adjoint optimization?

Our reply: Thanks for pointing out this important issue. The “target” of our genetic algorithm refers to the optimization convergence. When the improvement of the global fitness function value, i.e., figure of merit (FOM), is less than 0.000001 within 200 generations (2000 solutions) of iterations, the optimization process is judged to be converged and the pattern of the metasurface at this time is output as the final design. The FOM as a function of iterations during the process of optimization is shown in **Fig. S10**.

As for the choice of optimization method, the adjoint optimization method can also fulfill a similar optimization result, such as the paper titled “Multifunctional volumetric meta-optics for color and polarization image sensors” by researchers from California Institute of Technology (**Optica** 7, 280-283 (2020)), in which a 3D multilayers nanostructures is theoretically designed to realize the RGB color router. However, it should be noted that the RGB light distributions on the pixels in that work are fixed to be points, which enables to complete the backward adjoint simulation process of adjoint optimization. In contrast, the design goal of our work is to ensure as much RGB light as possible captured by the corresponding pixels in the imaging surface, and there is no restriction on the light distribution pattern on the pixels. If we impose some requirement on the RGB light distribution on the pixels, the designed color router will be limited and the optimal efficiencies will not be achieved (see **Fig. r3**, this issue will also be discussed in the response to Question 4), that is not our purpose. Moreover, the adjoint optimization is a local optimization algorithm, while genetic algorithm is a global optimization algorithm. Therefore, considering all the above reasons, we choose the genetic algorithm in this work.

Fig. s10. FOM as a function of iterations during the process of optimization. Insets: Pattern examples generated during optimization.

Fig. r3. The optimized design by setting smaller monitoring windows to show good focusing field patterns. The solid lines in the spectral efficiency diagram represent the efficiencies of color router (Subscript 0) reported in our paper, while the dotted lines represent the efficiencies of the color router (Subscript 1) with good focusing field patterns.

3. A potential issue with the style of design yielded by the inverse design algorithm is its highly 'pixelated' nature. How close are the fabricated structures to the inverse-design metasurface? Relatedly, how does the color filter performance compare to the Numerical simulations? There is no simulation data as far as I can see.

Our reply: In the fabrication of the metasurface sample, the period of the fabricated structures can be strictly controlled and the central positions of every nano-pillars can keep accurate, but the side lengths of the nano-pillars will be enlarged slightly. An SEM image of the metasurface sample is presented in **Fig. s12a**, which shows that the fabrication error is about 12% - 20% ($a=140\text{ nm} \sim 150\text{ nm}$). The efficiencies of color router with dilation of such side length have been calculated and shown in **Fig. s12b**. It is evident that the color collection efficiencies have the similar profiles and the peak efficiencies of these three color light are all larger than 50%, with different side lengths of nano-pillars. The measured peak efficiencies of the MBCR are 58%, 59% and 49% at the wavelengths of 640 nm, 540 nm, 460 nm, respectively, which is close to that of the simulation results (dash lines) in **Fig. s12b**. The experiment and simulation results have been shown as follows, which indicates that our metasurface color router is robust to the fabrication errors.

The discussions on this issue have been added to the Section 3 (the third paragraph) of the Supplementary Information.

Fig. s12. (a) Measurement of fabricated sample by proportional scaling. (b) Comparison between experiment and simulation (with variable side lengths of the pillars) results of RGB collection efficiencies in the visible region.

4. The performance of the green filtering seems quite poor to me. While blue and red seem fine, if I look at Fig. S3, the green power distribution seems almost random. This can be seen in Fig. 3F. I am surprised the authors state that there is little crosstalk issue when the power efficiency at say 500 nm is 50% for the green but is 0.4 and 650 and 700nm. This suggests to me the design is not nearly optimal for green - in fact the design seems to have optimized for good blue and red discrimination, but for green is fairly randomized.

Our reply: We thank the reviewer for pointing out this issue. In our optimization calculation, we do not impose any restriction on the light distribution on the corresponding pixels, but just adjust the weight of the function value of each color to pursue the highest average color collection efficiency and make the light energy captured by the corresponding pixels as much as possible, which is the reason why the pattern of green light is randomized. In fact, the green light distribution can be tuned to be more regular or well-focused by setting the calculation window of the green pixel to the required pattern. However, if we try to put some requirement on the distribution of green light, it will lead to low efficiencies. As a proof, we design a color router with the RGB light well-focused on centers of the corresponding pixels, shown in **Fig.r3**. It can be seen that the RGB light is well focused to the corresponding pixel centers, but the average color collection efficiencies of RGB (about 44%) and energy utilization efficiency are less than that of the structure we present in our paper. We do admit that with a better setting of fitness function (that considers both efficiency and good field patterns) and more optimization attempts, a solution that has a good tradeoff between high efficiency and good field patterns could be found.

Fig. r3. The optimized design by setting smaller monitoring windows to show good focusing field patterns. The solid lines in the spectral efficiency diagram represent the efficiencies of color router (Subscript 0) reported in our paper, while the dotted lines represent the efficiencies of the color router (Subscript 1) with good focusing field patterns.

5. What is the angular response of this metasurface filter? Presumably all the measurements have been at exactly normal incidence. But a traditional BCF does have reasonable angular acceptance range. I suspect this metasurface would not perform well off-normal. Either simulations or experimental data would be important to have here to understand the limitations of the metasurface.

Our reply: Thanks for pointing out this important issue. The angular response for oblique incidence has been numerically simulated and is shown in **Fig. s9**. It is seen that with the increasing of the incident angle, the peak values of all color collection efficiencies decrease gradually. Numerical simulation gives a maximum acceptable incident angle of about 30° , with the color routing function remaining. Under larger incident angles (40° and 50°), serious crosstalk occurs between RGB pixels and the color routing function cannot be realized. The imaging performance under off-normal incidence has also been experimentally studied, which gives a maximal numerical aperture (NA) value of 0.19 corresponding to an incident angle of 11° , as shown in **Fig. s22**. Although the color contrast slightly decreases and the image details are gradually lost, with the NA value

increasing, the quality of color imaging keeps quite good with the shapes and colors of the objects clearly distinguished, which is attributed to the high color collection efficiencies and energy utilization efficiency of MBCR. All these results demonstrate the robustness of our MBCR to oblique incidence.

The discussions on this issue have been added in the Results section (the fourth paragraph and eighth paragraph) of the manuscript.

Fig. s9. Transmittance curves of our MBCR when considering the angle of incidence up to 50° .

Fig. s22. Influence of numerical aperture (NA) on imaging performance. (a) Images taken at different numerical aperture (NA) of 0.02, 0.05, 0.09, 0.14, 0.19. (b) Image reconstruction of the pictures in (a).

6. Finally, there are quite a few grammatical errors and awkward phrasings in the manuscript. It could benefit from a closer reading/ editing for this.

Our reply: We thank the reviewer for this kindly suggestion. The manuscript is re-organized and extensively smoothed.

To Reviewer 3:

Instead of traditional filtering, the authors introduced the method of routing R, G, B colors into different regions, therefore, increases efficiency. The structure is only single-layer, and is designed using the GA optimization. Experiments demonstrated a 84% color collection efficiency over visible band. The designed structures can be directly applied to current Bayer-type image sensor to solve the low efficiency problem. The manuscript can be accepted with revision if the following issues are well-addressed.

Our reply: We thank the reviewer for the deep understanding of our work and the positive comments on it. We have addressed all the concerns raised by the reviewer, and revised the manuscript accordingly.

1. At line 102, the author mentioned the genetic algorithm is used for optimization. There are also other optimization methods, e.g., particle swarm, differential evolution, Bayesian optimization. Please explain why authors specifically choose the genetic algorithm.

Our reply: Genetic algorithm, particle swarm optimization, differential evolution, and adjoint optimization are in essence heuristic search algorithms, which has no advantages or disadvantages among them in principle. The optimized results can also be obtained by using other heuristic search algorithms, such as particle swarm optimization, differential evolution and so on. However, particle swarm optimization is not a global convergence algorithm and it is hard to set initial parameters because some empirical parameters need to be input before optimization. Similar with genetic algorithm, differential evolution algorithm randomly generates the initial population and takes the fitness value of each individual in the population as the selection standard. The main process also includes three steps: mutation, crossover and selection. The difference is that the mutation vector is generated by the parent's difference vector, and cross with the parent's individual vector to generate a new individual vector, and then be selected with its parent's individual. Bayesian optimization is a sequential design strategy for global optimization of black-box functions that does not assume any functional forms. It is usually employed to optimize expensive-to-evaluate functions, which is not suitable for this work because the objective function can be expressed quantitatively.

Furthermore, the adjoint optimization method can also fulfill a similar optimization result, such as the paper titled "Multifunctional volumetric meta-optics for color and polarization image sensors"

by researchers from California Institute of Technology (**Optica** 7, 280-283 (2020)), in which a 3D multilayers nanostructures is theoretically designed to realize the RGB color router. However, it should be noted that the RGB light distributions on the pixels in that work are fixed to be points, that enables to complete the backward adjoint simulation process of adjoint optimization. In contrast, the design goal of our work is to ensure as much RGB light as possible captured by the corresponding pixel in the imaging surface, so there is no restriction on the light distribution pattern on the pixels. If we impose any requirement on the RGB light distribution on the pixels, the designed color router will be limited and the optimal efficiencies will not be achieved, that is not our purpose. Moreover, the adjoint optimization is a local optimization algorithm, while genetic algorithm is a global optimization algorithm.

Therefore, considering all the above reasons, the genetic algorithm is the most suitable method for our work.

2. The genetic algorithm and the design process are not well discussed. Some information should be implemented or added:

(a) Fig. 2b seems too simplified. It is better to give more details on how GA works.

Our reply: Thanks for raising this point. Detailed information of how GA works has been added to the **Results** section (the third paragraph) of the manuscript, presenting as follows:

The flow diagram of the optimization algorithm shown in Fig. 2b is to clearly reveal several fundamental steps. First, the binary-pattern metasurface is encoded into the chromosome with binary numbers defined as either material (“1”) or free space (“0”). Second, to enable the optimization toward the optimal result, the fitness function that evaluates the routing property of incident light for each binary-pattern metasurface, is defined as

$$F = a_R \int_{\lambda_{R1}}^{\lambda_{R2}} T_R d\lambda + a_G \int_{\lambda_{G1}}^{\lambda_{G2}} T_G d\lambda + a_B \int_{\lambda_{B1}}^{\lambda_{B2}} T_B d\lambda,$$

where λ_{C1} and λ_{C2} are used to represent the minimum and maximum wavelengths and the subscript C represents R, G, and B color light. The minimum and maximum wavelengths are 600 nm and 700 nm for R, 500 nm and 600 nm for G, and 400 nm and 500 nm for B. In the above formula, T_R, T_G, T_B are the ratios of the light intensity of each color region at the imaging plane

to that incident on the unit cell, and a_R, a_G, a_B are the weights of the integral for each color. Thus, the design problem of a binary-pattern metasurface is formulated as maximization of the fitness function F . The GA is employed in combination with finite-difference time-domain simulations (Lumerical FDTD solutions) to optimize arrangement of the meta-atoms of the metasurface. Periodic boundary conditions are applied around the unit cell in simulation using FDTD software. The fitness value F is then calculated according to the transmission spectra of each obtained binary-pattern metasurface extracted from FDTD and is further evaluated to determine whether it has reached the target value. This optimization process continues until the GA stop criterion is met. Here, the stop criterion depends on the improvement of the global fitness function value is less than 0.000001 within 200 generations (2000 solutions) of iterations.

(b) The objective of the fitness function at line 124 should be discussed. At line 96-97, the authors mention “the routing property of the MBCR is adopted to replace both the XXXXX”. How this routing is reflected in the fitness function? How routing is adopted to replace XXX? Please add some discussions.

Our reply: Thanks for raising this point. The micro-lens above the color filter in traditional imaging system is to focus the incident light with a certain angle range to the color filter and is necessary for enhancing light harvesting efficiency. The color filter is responsible for transmitting light with specific wavelengths because the photodetectors are unable to distinguish the color. In the fitness function, these variables (T_R, T_{G1}, T_{G2} and T_B) are the transmittances of the four monitors calculated by solving the Maxwell equations in FDTD. The four monitors represent four different pixels detected by photodetectors. The fitness function aims to maximize the specific color light energy (wavelength range: $\lambda_{C1} \sim \lambda_{C2}$) within different color pixel areas, which is a routing process that makes incident light of different wavebands be transmitted to each pixel in a directional manner. *The discussions on this issue have been added in the Results section (the third paragraph) of the manuscript.*

Our color router with a unit cell area of $2 \times 2 \mu\text{m}^2$ can directly make the different color light “focus” into a smaller area of $1 \times 1 \mu\text{m}^2$, which illustrates that our color router can provide the function of “focusing”. This routing progress enhances light harvesting efficiency effectively and

sorts the incident light to RGB light, which is a functional upgrade of the filters. Moreover, the color router can concentrate the RGB light to the corresponding pixels under oblique incidence (0~30°). Therefore, we think the color router can replace both the color filters and micro-lens. We have modified the sentence as “*the routing property of the MBCR is adopted to improve the filtering of CFs and replace the focusing of the micro-lens in the common BCF*” in the **Results** section (the first paragraph) of the manuscript.

(c) Some hyperparameters as well as the computer resources of GA algorithms can be provided for referencing. Some intermediate results during GA algorithms can be provided, e.g., the fitness function vs. epoch during optimization, several examples of generated structures during optimization.

Our reply: Thanks for raising this point. Specifications of the workstations used for the numerical optimization and some other details of the optimization processes are set as follows:

- a) CPU: Intel Xeon Gold 6134 CPU @ 3.20 GHz
- b) RAM: 128 GB
- c) Meshsize: 20 nm (late stage) & 50 nm (early stage)
- d) Simulation time: 20 nm mesh: about 5 minutes 45 seconds per simulation (for efficiency >0.45); 50 nm mesh: about 55 seconds per simulation (for efficiency <0.45)
- e) Boundary conditions: X & Y directions: Periodic; Z direction: Perfect matching layer (PML)

The initial solutions are randomly generated with filling factor of 0.1. The number of solutions for each generation is set to be 10. The figure of merit (FOM) line is shown in **Fig. s10**. The horizontal axis in **Fig. s10** is the generation numbers of iteration. The solid red line shows how the global best FOM evolves with iterations. The dashed gray line shows how the best FOM of each generation evolves with iterations. Parts of the specific structures for the best global FOMs are also show in the insets, from which we can see how the structure evolves as the FOM increases.

Fig. s10. FOM as a function of iterations during the process of optimization. Inset: The pattern examples generated during optimization.

(d) More illustrations of why this structure can give the routing ability is needed, which can help researchers to understand the physics. Some pointing vector distributions are given in In Fig. 2c-e. However, this is not sufficient. It would be better to also provide the power flow in the z direction/ in 3d, and the electrical field at the designed structures.

Our reply: Thanks for raising this point. In order to understand the physical mechanism of light routing for the optimized pattern in our paper, optical mode analysis is carried out at the three peak wavelengths ($\lambda = 450$ nm, 540 nm, and 660 nm). **Figure s5** shows the top view of electric field $|E|$ distributions of one unit cell of the optimized color router. It shows that the electric field distributions are enhanced inside the nanostructures of the metasurface for shorter wavelengths, while for longer wavelength enhanced in the gaps between nanostructures. Moreover, we added the power flow $|P|$ maps (**Fig. s4**) in the z direction at $\lambda = 450$ nm, 540 nm, and 660 nm, which is consistent with the top view of Poynting vector maps in the manuscript of **Fig. 2c-e** and light is effectively routed to the corresponding pixels.

Descriptions on this issue are added in **Section 1** (the second and third paragraphs) of the **Supplementary Information**, as follows:

We also plot the power flow $|P|$ maps in the z direction at $\lambda = 660$ nm, 540 nm, and 450 nm shown

in Fig. s4, which is consistent with the top view of Poynting vector maps in the manuscript of Fig. 2c-e and light is effectively routed to the corresponding pixels. In order to understand the physical mechanism of light routing for the optimized pattern, optical mode analysis is conducted at the three peak wavelengths ($\lambda = 450$ nm, 540 nm, and 660 nm), as shown in Fig. s5a-c. It shows that the electric field $|E|$ distributions are enhanced inside the nanostructures of the metasurface for shorter wavelengths, while for longer wavelength enhanced in the gaps of nanostructures.

Fig. s5. Top view of the electric field $|E|$ distributions in the metasurface at $\lambda = 450$ nm, 540 nm, and 660 nm.

Fig. s4. Cross-sectional view (XZ plane) of the pointing vector $|P|$ distributions in the z direction at $\lambda = 660$ nm, 540 nm, and 450 nm at the locations of $y = -0.6$ μm , 0.2 μm , 0.5 μm . The black dash lines represent the interface between Si_3N_4 metasurface and quartz substrate or the interface of different

color pixels in Z direction.

(e) In Fig. 2d, there are some crosstalk for the green light. I wonder if this is because of the fitness function, where you want to distribute green light to two different regions. Will the crosstalk decrease if you only let the green light to route into one region? In terms of real application, there is no need to exactly follow the light distribution of Bayer-type image sensor, as long as they are compatible.

Our reply: We thank the reviewer for this kindly suggestion. If the green light is routed into one region, (that is, the pixel positions of red, green and blue are arranged according to the law of wavelength dispersion), the crosstalk will be easily reduced. In our work, we use the Bayer-type pattern for the compatibility of the metasurface with the existing de-mosaic algorithm to get colorful images and easy comparison with traditional Bayer color filters. In future, we will consider designing non-Bayer-type color router to further promote the development of the miniaturized imaging systems. It is not a big challenge but only requires some computation time for our current algorithm to design a metasurface for other color distributions. Some preliminary results for RGB distribution simply aligning horizontally are shown in **Fig. r4** as follows.

Fig. r4. Design of non-Bayer-type color router with RGB distribution simply aligning horizontally.

The solid lines in the spectral efficiency diagram represent the efficiencies of RGG-pattern color

router (Subscript 0) reported in our paper, while the dotted lines represent the efficiencies of RGB-pattern color router (Subscript 2).

Some other minor comments:

1. At line 47-49, the reason why using metallic plasmonic structures or photonic crystal structures can improve the transmission can be briefly discussed.

Our reply: Different from the traditional pigment/dye-based color filters work based on the wavelength-selective optical absorption/reflection of the chemical bonds, metallic plasmonic structures or photonic crystal structures realize the function of filtering due to the interaction between incident light and nanostructures. The enhanced transmission in metallic plasmonic structures can be explained by excitation of surface plasmon polaritons. Transmission enhancements usually determined by various parameters of the holes and thickness of metal film, as well as the optical properties of metal and dielectric medium. In contrary to metallic plasmonic materials, photonic crystal structures based on all-dielectric materials provide a lossless optical system and 2D periodic nanostructures relying on an electric dipole and a magnetic dipole, which are mediated by the Mie scattering, are employed to enhance the wavelength-dependent optical transmission.

2. From line 53-79, many discussions are not necessary. Please delete some redundant information to make it concise. Similar issues can be found at line 97-101, where the inverse design was already mentioned in the introduction.

Our reply: Great thanks for your suggestion. We have already deleted the redundant sentences mentioned above. Additionally, the whole manuscript has been checked further and revised.

3. In Fig. 2c-e, the calculated Poynting vector distributions are provided to visualize the power flow density distributions. The blue one is hard to see in the dark background, is it better to view in white background? Also, what is the light distribution slightly outside the color pixel? Is there any color crosstalk at nearby effective color pixel?

Our reply: We thank the reviewer for this kindly suggestion. The Poynting vector distribution maps with white background also looks low-contrast, so we use another color bar with high color contrast and replace the original picture with it. There is some crosstalk at nearby effective color pixel. In

addition to the crosstalk of green light, the crosstalk of red and blue light can be almost ignored.

4. From line 115-117, I am confused what the authors refer to for Fig. 2(a). Is this the structure at the image plane, or this is the metasurface? Please be clear.

Our reply: Thank for your comments and suggestions. In fact, this is one unit cell of metasurface corresponding to 4 pixels (marked by different color dotted lines) on the imaging plane. We additionally mark the regions of 4 pixels with the aiming to easily understand the spatial correspondence between the unit cell and the pixels. According to your suggestion, we have added some detailed descriptions in the **Results** (the second paragraph) of the main manuscript so as not to cause some confusions in understanding.

5. At line 170, the author mentioned “The RGB lights are well routed and focused into their target quadrants”. It would be better to describe where the focusing happens.

Our reply: Thanks for raising this point. Our color router with a unit cell area of $2 \times 2 \mu\text{m}^2$ can directly make the different color light “focus” into a smaller area of $1 \times 1 \mu\text{m}^2$, which illustrates that our color router can provide the function of “focusing”. This routing progress enhances light harvesting efficiency effectively and sorts the incident light to RGB light, which is a functional upgrade of the filters. Moreover, the color router can concentrate the RGB light to the corresponding pixels under oblique incidence ($0 \sim 30^\circ$). Therefore, we think the color router can replace both the color filters and micro-lens. We have describe this point with “*the routing property of the MBCR is adopted to improve the filtering of CFs and replace the focusing of the micro-lens in the common BCF*” in the **Results** section (the first paragraph) of the manuscript.

6. At line 165, and also 172-173, the author mention the field distribution is given in Fig. 3e. Field distribution usually contains both amplitude and phase, here is only the intensity of E field.

Our reply: Field distribution here represents the intensity of light distribution in the focal plane, which corresponds to the magnitude square of E field. We don't care about the specific phase value here. *We replace the expression of 'field distribution' in the manuscript with 'the light intensity distribution on the imaging plane' to avoid misunderstanding.*

7. The method of reconstruction of colorful image at line 195-198 is not clearly stated.

Our reply: Thanks for raising this point. More detailed descriptions as follows have been added to **Results** (the seventh paragraph) of the manuscript.

Figure 4a schematically shows the post-processing procedure of color imaging using the MBCR. A raw grey picture with a mosaic pattern is directly captured by the MBCR and monochromatic imaging sensor. Then, the spectral responses of the MBCR are imposed on each pixel to obtain the color figure with a mosaic pattern using the conversion matrix method. The conversion matrix directly converts the detected intensity of spectral values into three-channel RGB values. The final step is to demosaic the color figure with a mosaic pattern. That is, according to the above RGGB information, the pixel interpolation algorithm (e.g., demosaicing interpolation) is carried out to obtain the RGB values of each RGGB unit cell. The colorful picture is ultimately reconstructed by transforming the RGGB pixel values to one three-channel pixel value, simulating the three-channel imaging in the actual color imaging sensor.

8. Fig. S4 and S5 are not compatible, there is no total efficiency in S5 for comparison.

Our reply: We thank the reviewer for this kindly suggestion. We have added the total efficiency in Fig. S5 for comparison.

Fig. s6. The color collection efficiencies calculated by the transmittance curves of commercial Bayer CFs (see Ref. 48 in main text).

9. Fig. S11, some figures are too dark too be seen. Maybe improving the brightness or contrast will help?

Our reply: We thank the reviewer for this kindly suggestion. We have zoomed in on the details of the image and improved the quality of these images.

Fig. s15. Measured light intensity distribution on the imaging plane under different wavelength of incident light in linear scale with the MBCR. The wavelength of the incident light is labeled above each picture.

REVIEWERS' COMMENTS

Reviewer #1 (Remarks to the Author):

I am pleased to say that the authors have satisfactorily addressed all my previous concerns. I am in the pleasant position to now recommend this article for acceptance pending a final proof editing.

Reviewer #2 (Remarks to the Author):

The authors have largely addressed my concerns. I think one point remains that I am still a bit unclear about, but I do not need to see the revision for this - just something that may help the manuscript further:

Crosstalk and color matching remain the likely biggest weakness of the presented data. I think it should be possible to define a different objective function for the genetic algorithm around color fidelity (or perhaps adding it to the objective along with spectral throughput) that would reduce crosstalk. Is there some fundamental limit here? For example, Fig. R3 shows a design with cleaner green focusing, but also significant crosstalk of green light to the red detector. Ultimately, for adoption as a single layer metasurface design, solving the color fidelity issue will be key. It is certainly not expected that it should be solved in this paper - however some further discussion to this point would nicely frame the results, and highlight the next steps.

Reviewer #3 (Remarks to the Author):

I am satisfied with students response and revised manuscript, and recommend acceptance.